# Light-driven decarboxylative deuteration enabled by a divergently engineered photodecarboxylase

Jian Xu[1,2,4 ✉], Jiajie Fan[1,4], Yujiao Lou[1,4], Weihua Xu[1,4], Zhiguo Wang [3,4], Danyang Li[1], Haonan Zhou[1], Xianfu Lin[1] & Qi Wu [1 ✉]

Despite the well-established chemical processes for C-D bond formation, the toolbox of enzymatic methodologies for deuterium incorporation has remained underdeveloped. Here we describe a photodecarboxylase from *Chlorella variabilis* NC64A (*Cv*FAP)-catalyzed approach for the decarboxylative deuteration of various carboxylic acids by employing $D_2O$ as a cheap and readily available deuterium source. Divergent protein engineering of WT-*Cv*FAP is implemented using Focused Rational Iterative Site-specific Mutagenesis (FRISM) as a strategy for expanding the substrate scope. Using specific mutants, several series of substrates including different chain length acids, racemic substrates as well as bulky cyclic acids are successfully converted into the deuterated products (>40 examples). In many cases WT-*Cv*FAP fails completely. This approach also enables the enantiocomplementary kinetic resolution of racemic acids to afford chiral deuterated products, which can hardly be accomplished by existing methods. MD simulations explain the results of improved catalytic activity and stereoselectivity of WT *Cv*FAP and mutants.

[1] Center of Chemistry for Frontier Technologies, Department of Chemistry, Zhejiang University, Hangzhou, P. R. China. [2] College of Biotechnology and Bioengineering, Zhejiang University of Technology, Hangzhou, P. R. China. [3] Institute of Aging Research, School of Medicine, Hangzhou Normal University, Hangzhou, P. R. China. [4] These authors contributed equally: Jian Xu, Jiajie Fan, Yujiao Lou, Weihua Xu, Zhiguo Wang. ✉email: jianxu@zjut.edu.cn; wuqi1000@163.com

Deuterium labeling is an attractive and powerful tool in the investigation of metabonomics, proteomics, mass spectrometry studies and reaction mechanisms[1–3]. In pharmaceutical chemistry, the introduction of C–D bonds has recently aroused scientific interests as an effective approach to modify the absorption, distribution, and toxicological properties of therapeutic drug molecules, which is due to the higher chemical inertness of C–D bonds compared with the C–H bonds[4]. Moreover, Austedo™, the first deuterated drug approved in 2017 by the US Food and Drug Administration, has been a major driving force for the booming development of the research of deuterium incorporation[5].

A number of elegant chemical methods for deuteration have been exploited over the past decades[6–11]. However, most of these approaches rely on transition metal complexes or harsh reaction conditions, which may suffer from high costs and raise environmental issues. For example, highly active Hydrogen Isotope Exchange (HIE) reactions by heterogeneous approaches has been found with palladium, platinum, rhodium, nickel, and ruthenium catalysts[11]. Moreover, among all transition metals employed in homogeneous HIE reactions, iridium is arguably the most widely studied[8,9,11]. Recently, photocatalytic deuteration of synthetically valuable organic molecules through mild routes has received considerable attentions[12,13]. MacMillan's group developed a strategy for direct deuteration of pharmaceutical compounds with a photoredox-hydrogen atom transfer-catalyzed process in equilibrium with $D_2O$[14]. Typically, heterogeneous metal-catalyzed HIE as well as some photocatalytic deuterations results in relatively unspecific incorporation of numerous deuterium atoms into a molecular substrate[11,13]. Accordingly, there is a great need for the development of mild and selective methodologies for the incorporation of a single deuterium. Deuterodefunctionalization has recently found widespread application in the selective incorporation of deuterium[15], and a few kinds of deuterodefunctionalization transformations have been developed, such as deborylative[14], decarboxylative[16–18], deoxygenative[19], and dehalogenative[20–22] deuteration. Among them, protodecarboxylation reaction is of high preparative utility in this field, first as a constructive model method for selective deuteration and also as a precursor step of decarboxylative cross-coupling reaction for regiospecific C − C and C − heteroatom bond formation[23,24]. In these processes, precious metals such as palladium, silver, were principally used as catalysts, although some copper-catalyzed deuterodecarboxylations[16] were also reported as exceptional cases, while they usually hampered by the extremely high temperature and costly ligands[23]. Recently, some mild deuterium exchange reaction of free carboxylic acids by photochemical decarboxylation have also been successfully demonstrated[25,26].

Biocatalysis has emerged as a powerful tool in organic synthetic chemistry because enzymes generally display high activity and selectivity under mild reaction conditions[27–30]. Although the chemical deuteration methods have been well studied, the toolbox of enzymatic methodologies for deuterium incorporation has remained underdeveloped[31,32]. To the best of our knowledge, there are mainly two kinds of enzymatic deuteration methodologies reported so far. Biocatalytic reductive deuteration[31,33–38] requires a supply of deuterated, reduced cofactor, [4-D]-NADH, which must be continually regenerated in situ by the appropriate dehydrogenase enzyme, in conjunction with a super-stoichiometric supply of a sacrificial deuterated reductant, D-ethanol, D-isopropanol, D-glucose or D-formate[33,34]. More recently, several works successfully demonstrated $H_2$-driven [4-D]-NADH recycling using hydrogenase and NAD+ reductase, with $D_2O$ supplying the deuterium atoms[31,36,37]. Various NADH-dependent reductases such as ADH (Alcohol dehydrogenase)[31,36]

and IRED (Imine reductase)[37] can provide corresponding deuterated chiral alcohols and amines. Another kind of enzymatic deuteration methodology was deuterodecarboxylations catalyzed by aromatic L-amino acid decarboxylase, mainly reported by Kańska group[39–41]. This method can prepare specific deuterated aromatic amines starting from various aromatic L-amino acids such as L-tryptophan, L-phenylalanine, L-tyrosine, and their derivatives by applying the appropriate aromatic L-amino acid decarboxylase in deuterated media. Accordingly, the substrate scope of these deuterodecarboxylations catalyzed by aromatic L-amino acid decarboxylase is very narrow. Indeed, biocatalytic aliphatic decarboxylative deuterations providing deuterated alkanes have never been reported.

Recently, photoenzymes, which directly utilize visible light to activate catalytic activity, have been exploited to generate new carbon skeletons, resulting in the diversification of natural products with improved bioactivities[42]. For example, Hyster's group has utilized photoexcited natural enzymes to be competent for new transformations, including reduction, dehalogenation, deacetoxylation or even cyclization[43–48]. Fatty acid photodecarboxylase from *Chlorella variabilis* NC64A (*Cv*FAP) is another type of photoenzymes discovered by Beisson and co-workers[49]. In other key studies, Hollmann's group has brought *Cv*FAP into chemistry by designing cascade reactions or decoy molecules[50–52]. Scrutton's group has developed a strategy for the production of bio-alkane gas with engineered *Cv*FAP[53]. In conjunction with our efforts, the feasibility of kinetic resolution of α-functionalized carboxylic acids by engineered *Cv*FAP has been investigated[54]. In the proposed mechanism of *Cv*FAP-catalyzed decarboxylation, alkyl carboxylate is activated by single electron transfer (SET) with FAD* to give a carboxyl radical, which undergoes facile decarboxylation to yield the corresponding alkyl radical[55,56]. On this basis, we questioned whether we could exploit the alky radical to access deuterated products. We speculated that if $D_2O$ is present, the generation of active deuterium would subsequently interact with the alkyl radical, resulting in the formation of a C–D bond through hydrogen atom transfer (HAT) (Fig. 1a).

Herein, we report an enzymatic methodology that introduces deuterium into organic molecules with $D_2O$ as an easily handled, cheap, and readily available deuterium source under mild reaction conditions. Although the biocatalysts can hardly be "universal," we engineered *Cv*FAP by divergent pathways for a broad range of substrates. Using specific mutants, several series of substrates including different chain length acids, racemic substrates as well as bulky cyclic acids were successfully converted into the deuterated products.

## Results and discussion

**Reaction conditions**. We commenced our study by evaluating the viability for the decarboxylation deuterium incorporation with palmitic acid (**1A**, Table 1) as starting material because it is naturally occurring and has the highest decarboxylation activity under the catalysis of WT-*Cv*FAP[49,50]. After irradiation for 12 h with $D_2O$ as solvent, the reaction proceeded smoothly to give the desired product in 99% yield with 93% D-incorporation (Table 1, entry 1), demonstrating the feasibility of the hypothesis. Importantly, the use of free FAD cofactor gave no observable product, implying the necessity of the protein scaffold (Table 1, entry 2). As expected the use of DMSO or $CH_3CN$ as co-solvent to improve the substrate solubility resulted in a significant increase in efficiency (Table 1, entry 5, 6). Further control experiments revealed the requirement of *Cv*FAP and a light source (Table 1, entry 3, 4). Moreover, in contrast to traditional chemical photoredox deuteration, we discovered that this bio-radical

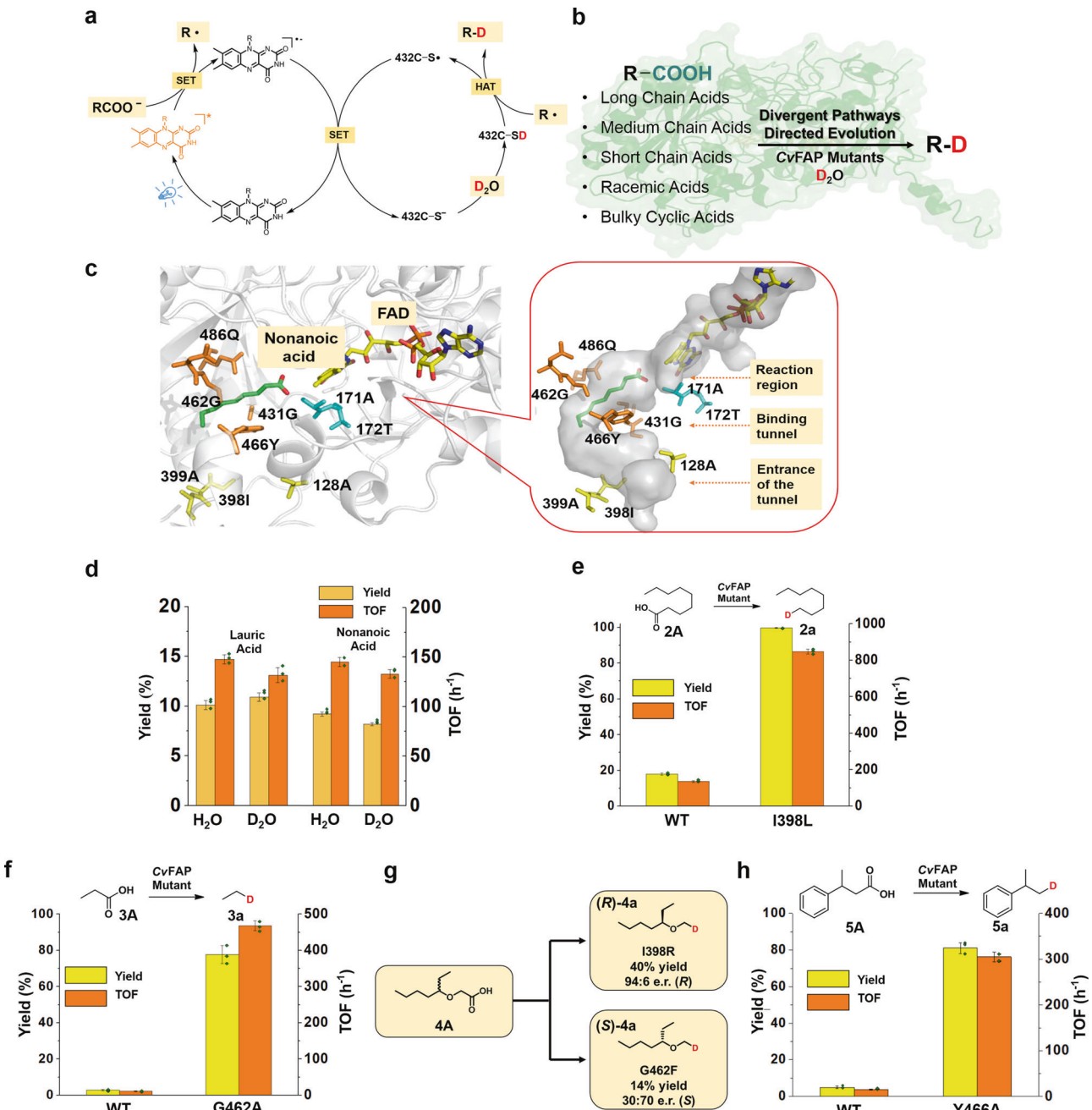

**Fig. 1 Design of *CvFAP*-catalyzed decarboxylative deuteration and divergent directed evolution of *CvFAP*. a** Proposed mechanism of *CvFAP*-catalyzed decarboxylative deuteration. **b** Design of divergent directed evolution of *CvFAP* toward various substrates. **c** Docking result with nonanoic acid (green) and selected hot positions for protein engineering (PDB:5NCC)[49], FAD (flavin adenine dinucleotide) and some representative residues located at the entrance of the substrate pocket (yellow), binding region (orange) and reaction region (cyan) are represented by sticks with different colors. **d** The influence of kinetic isotope effect on the reaction activity. **e** The comparison of yields and TOF between WT-*CvFAP* and the best mutant (I398L) for the medium chain acid. **f** The comparison of yields and TOF between WT-*CvFAP* and the best mutant (G462A) for the short chain acid. **g** The enantiodivergent decarboxylative deuteration catalyzed by I398R and G462F mutants, respectively. **h** The comparison of yields and TOF between WT-*CvFAP* and the best mutant (Y466A) for bulky cyclic acid. Error bars represent the mean ± SD of three independent experiments. Source data are provided as a Source Data file.

transformation can be run in a manner open to air and without the requirement of extra thiol as a hydrogen atom transfer (HAT) additive (Table 1, entry 8)[13] Lastly, the influence of the various cosolvent addition, substrate concentration and reaction time were further studied and the best result of WT *CvFAP*-catalyzed decarboxylation deuterium of palmitic acid was obtained from the reaction with 100 mM substrate concentration and 20% vol. DMSO as cosolvent after 12 h (Supplementary Figs. 1–3).

Moreover, the influencing factor for D-incorporation was also investigated. According to the reaction mechanism, we considered the major influencing factor for high D-incorporation is the concentration of residual $H_2O$ in the reaction system (Supplementary Fig. 4). The extent of labeling roughly follows the ratio of $D_2O$ in the solvent mixture, suggesting kinetic isotope effect is small. With the prolongation of freeze-drying time of enzymes and the continuous removal of $H_2O$, the D-incorporation

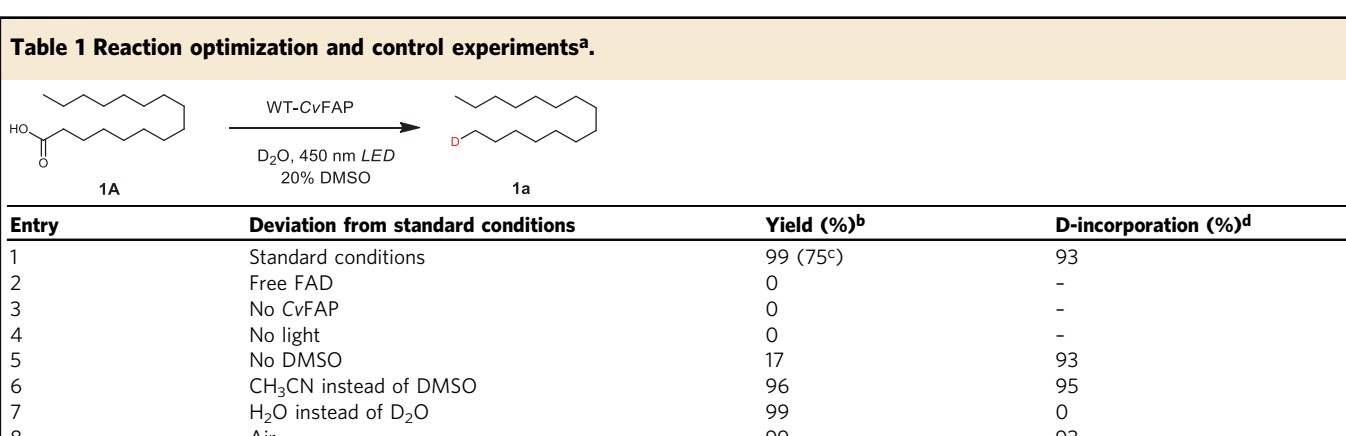

**Table 1 Reaction optimization and control experiments[a].**

| Entry | Deviation from standard conditions | Yield (%)[b] | D-incorporation (%)[d] |
|---|---|---|---|
| 1 | Standard conditions | 99 (75[c]) | 93 |
| 2 | Free FAD | 0 | – |
| 3 | No CvFAP | 0 | – |
| 4 | No light | 0 | – |
| 5 | No DMSO | 17 | 93 |
| 6 | CH₃CN instead of DMSO | 96 | 95 |
| 7 | H₂O instead of D₂O | 99 | 0 |
| 8 | Air | 99 | 93 |

[a]Reaction conditions: Substrate (0.40 mmol), crude enzyme powder (containing CvFAP about 20 mg), D₂O (4 mL), DMSO (1 mL), 450 nm LED, 20 °C, 12 h.
[b]Yields were determined by GC.
[c]Isolated yield.
[d]D-incorporation are determined by $^1$H NMR or HRMS.

improved significantly. Thus, we chose 10 h for the freeze-drying treatment of CvFAP before starting reactions, it is sufficient to ensure high D-incorporation.

**Scope of WT-CvFAP catalyzed decarboxylative deuteration.**
With the optimized conditions in hand, we surveyed the substrate scope of this bio-deuteration process. As depicted in Fig. 2, a wide range of differently substituted long chain fatty acids were readily converted into their corresponding deuterated compounds with high yields and excellent D-incorporation (Fig. 2, **1a–1k**) It is worth noting that introducing active groups such as −OH, −O−, or -alkenyl moieties on the carbon chain did not affect the reaction results adversely. Natural products such as oleic and linoleic acids were also suitable substrates, affording the corresponding products with good yields and excellent D-incorporations (**1d**, **1e**). Unfortunately, when switching the long chain acids to short, medium chain acids or aromatic acids, the products yields decreased significantly (Fig. 2, **2a**, **5a**). To overcome this limitation, we employed divergent directed evolution for CvFAP to access different functionalized variants for accepting a diversity of substrates.

**Divergent directed evolution of CvFAP for diverse substrates.**
We focused our initial directed evolution route on the deuteration reaction of medium chain fatty acids. Following a thorough analysis of the crystal structure of the CvFAP active site[49] into which we docked nonanoic acid as the model substrate (Fig. 1c), we divided the active site into three domains, the entrance of the substrate pocket, the hydrophobic binding region and the reaction region. The hot sites for mutation lining the binding and reaction region (CAST sites)[29,57] were selected to be the largest potential contributors in influencing the activity. Meanwhile, other sites at the entrance of the binding pocket which were relatively far from the substrate were also selected for mutagenesis due to the consideration of possible cooperative mutational effects. Since the kinetic isotope effect shows a small effect on the activity of CvFAP -catalyzed decarboxylation (Fig. 1d), and considering economical factors and operation convenience, we chose H₂O as solvent in the model reactions for subsequent screening of mutant libraries. We then proceeded with protein engineering under the guidance of "Focused Rational Iterative Site-specific Mutagenesis" (FRISM)[58,59], which constitutes an effective fusion of directed evolution and rational design. FRISM is an offspring of CAST/ISM[29,57], but it does not require the formation of mutant libraries generated by focused saturation

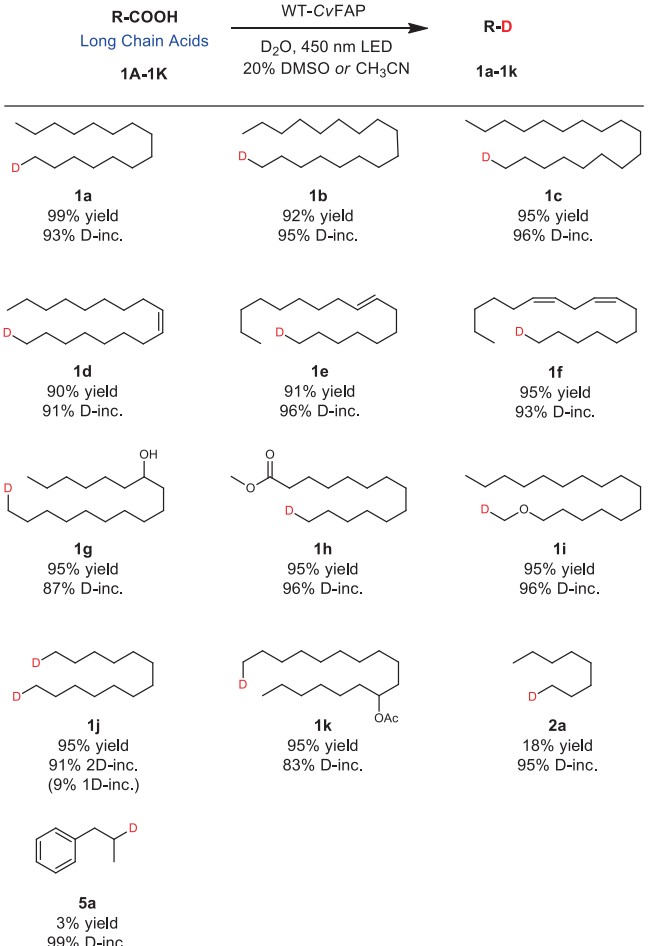

**Fig. 2 Deuteration scope: long chain acids.** Reaction conditions: Substrate (0.40 mmol), crude enzyme powder (containing CvFAP about 20 mg), D₂O (4 mL), DMSO or CH₃CN (1 mL), 450 nm LED (light-emitting diode), 20 °C, 12 h. Yields were determined by GC. D-inc. data were determined by $^1$H NMR or HRMS.

mutagenesis, and therefore circumvents laborious screening[54,60]. Accordingly, site-specific mutagenesis with the introduction of differently sized amino acids, including alanine (A), leucine (L), and phenylalanine (F), was first used to modify the volume of

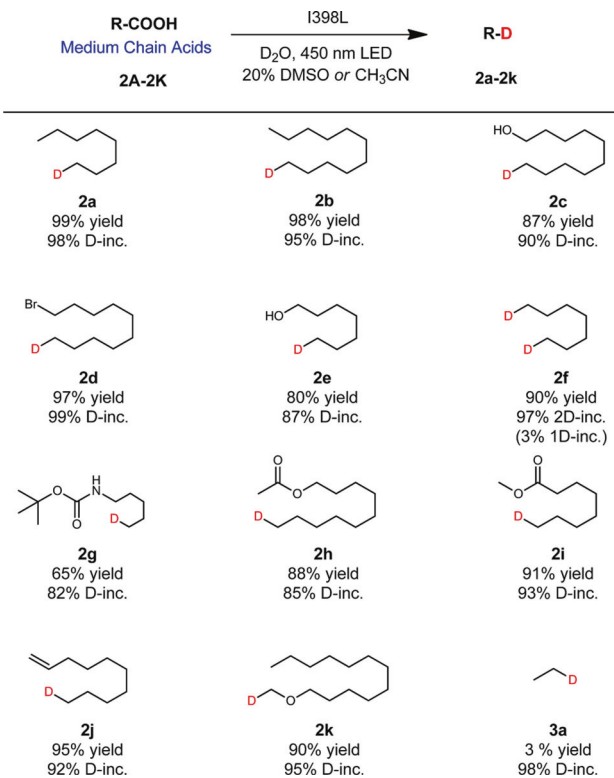

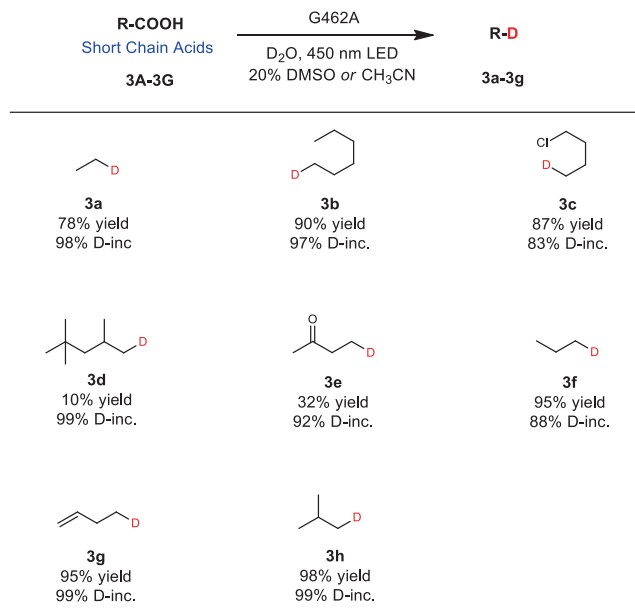

**Fig. 3 Deuteration scope: medium chain acids.** Reaction conditions: Substrate (0.40 mmol), crude enzyme powder (containing $Cv$FAP about 20 mg), $D_2O$ (8 mL), DMSO or $CH_3CN$ (2 mL), 450 nm LED, 20 °C, 12 h. Yields were determined by GC. D-inc. data were determined by [1]H NMR or HRMS.

**Fig. 4 Deuteration scope: short chain acids.** Reaction conditions: Substrate (0.40 mmol), crude enzyme powder (containing $Cv$FAP about 20 mg), $D_2O$ (4 mL), DMSO or $CH_3CN$ (1 mL), 450 nm LED, 20 °C, 12 h. Yields were determined by GC. D-inc. data were determined by [1]H NMR or HRMS.

these hot-spots. Afterwards, the best hits were then tested with similarly sized amino acids (A$^+$, L$^+$, or F$^+$) as an "extension-library" for possible activity improvement. A$^+$, L$^+$, and F$^+$ include G, V/C/I/M and Y/W, respectively. To our surprise, most tested mutations at the binding or reaction region displayed only slight or no influence on the reaction activity of nonanoic acid (Supplementary Fig. 5), while one beneficial mutation, I398L which is located at the entrance to the binding pocket, showed a nearly 10-fold improvement of activity, thereby furnishing a satisfactory yield and catalytic efficiency (Fig. 1e). Another positive mutation, A128L, is also at the entrance, and likewise showed a significant enhancement of activity (Yield 93%, TOF 725 h$^{-1}$).

Next, we investigated the scope of substrates using variant I398L, which is by far the best mutant (Fig. 3). To our satisfaction, the decarboxylative deuteration reactions of these tested medium chain acids derivatives were successfully realized under the catalysis of I398L in good yields and D-incorporation. Thus, promising functional group tolerance was accomplished by generating and screening less than 30 variants (Supplementary Fig. 5). Nonetheless, the catalytic activity of the I398L mutant decreased significantly when the carbon chain becomes shorter (Fig. 3, 3a), which prompted us to search a new mutant to accept short chain acids.

With these considerations in mind, propionic acid was used as the model substrate for the further divergent screening of FRISM mini-libraries. To our delight, mutant G462A furnished a notable improvement of yields (78%) and TOF (468 h$^{-1}$) (Fig. 1f). A number of short chain acids were then subjected to the best mutant, and indeed the corresponding deuterated products 3a−3g were obtained in high yields (Fig. 4). Particularly

noteworthy is that the gaseous deuterated compounds could also be obtained successfully by this method (Fig. 4, 3a, 3f–3h).

Generally, enzymes offer attractive stereoselectivity under mild conditions, complementing or replacing manufacturing processes based on traditional man-made catalysts. However, the development of an enzyme for enantioselective synthesis of these deuterated compounds remains a challenge. Based on our previous studies in the $Cv$FAP-catalyzed kinetic resolution of α-functionalized carboxylic acids[54], we envisioned that the chiral deuterated products could also be achieved by engineered $Cv$FAP. Since optically active ethers and carboxylic acids are important chiral building blocks in pharmaceutical chemistry, we chose 2-(heptan-3-yloxy) acetic acid (4A) as an ideal model substrate for screening (Fig. 1g). WT-$Cv$FAP was found to be selective for the ($R$)-configuration with an enantiomeric ratio (e.r.) of 95:5. However, the carbon chain is too short to be accepted by WT-$Cv$FAP, leading to low activity (yield 19%). The yield did not increase with prolongation of reaction time[51]. Upon screening the original $Cv$FAP library, mutant I398R was found to be effective. A 40% yield with satisfactory preference for the ($R$)-configuration (e.r. = 94:6) (Fig. 1g). Next, we wanted to see how the best mutant performs in the decarboxylative deuteration of other differently substituted carboxylic acids (Fig. 5). To our delight, a range of tested deuterated chiral ether products were obtained with good yields and stereoselectivity (4a–4i). However, due to the large structural difference, the activity and stereoselectivity of deuterated alcohol products decreased significantly (4j, 4k). Further studies on directed evolution of highly stereoselective $Cv$FAP for such compounds or other synthetically useful chiral substrates are ongoing in our laboratory.

Directed evolution of $Cv$FAP furnished a complementary set of enzymatic catalysts that allow for the enantiodivergent deuteration. Residue 462 G, which is located in the binding region as a type of CAST site[29,57], was identified as the key position related to stereoselectivity in this kinetic resolution[54]. The single G462F mutation inverted the absolute configuration of model substrate (4A) in favor of ($S$)-selectivity (Fig. 1g). We note that this is the first example of enantiodivergent evolution using $Cv$FAP.

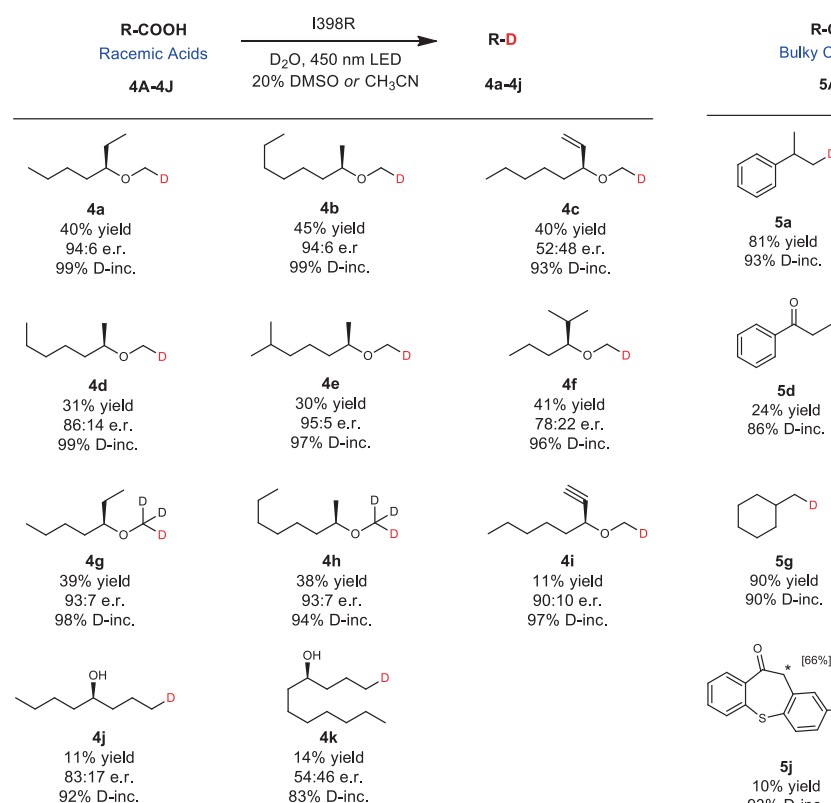

**Fig. 5 Deuteration scope: racemic acids.** Reaction conditions: Substrate (0.40 mmol), crude enzyme powder (containing $CvFAP$ about 20 mg), $D_2O$ (4 mL), DMSO or $CH_3CN$ (1 mL), 450 nm LED, 20 °C, 12 h. Yields and e.r. values were determined by chiral GC. D-inc. data were determined by $^1H$ NMR or HRMS. E.r. value of **4e** was calculated based on the conversion and e.r. of **4E**.

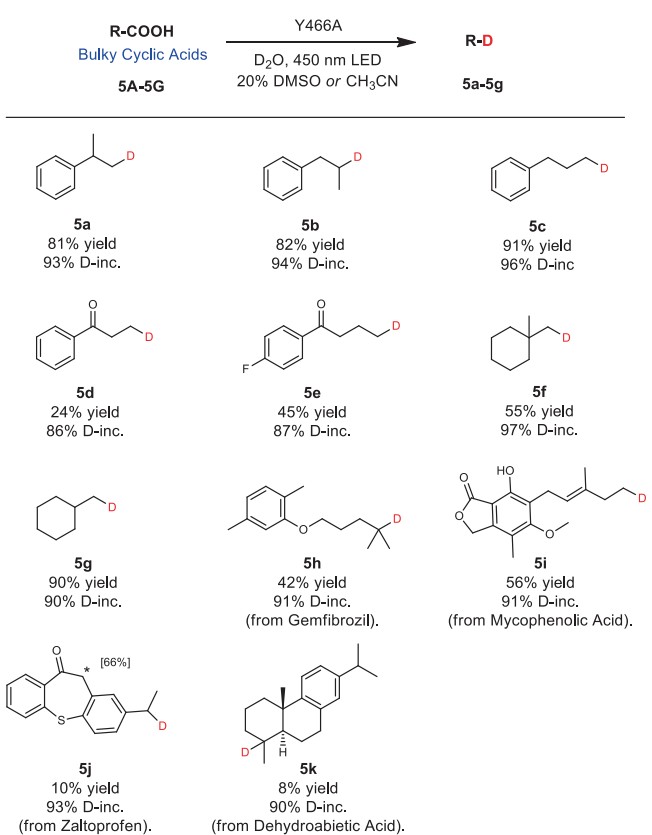

**Fig. 6 Deuteration scope: bulky cyclic acids.** Reaction conditions: Substrate (0.40 mmol), crude enzyme powder (containing $CvFAP$ about 20 mg), $D_2O$ (4 mL), DMSO or $CH_3CN$ (1 mL), 450 nm LED, 20 °C, 12 h. Yields were determined by GC. D-inc. data were determined by $^1H$ NMR or HRMS. *Non-enzymatic D-incorporation at C11 of **5j** was also detected (see Supplementary Fig. 87).

Next, in order to improve activity in the reaction of bulky cyclic compounds, a final evolution route of $CvFAP$ was explored. Residue Y466 was considered to play a dominant role in bulky substrate recognition, because it occupies a large space in the narrow tunnel near the active site, resulting in the stabilization of chain substrates by hydrophobic interactions[49]. Thus, Y466 was subjected individually to site-specific mutagenesis with the introduction of alanine (A) and glycine (G) mutations to enlarge the binding tunnel, hopefully facilitating better substrate binding at the active site. Indeed, mutant Y466A induced activity significantly for 3-phenylbutanoic acid (**5A**) relative to WT-$CvFAP$ (Figs. 1h and 6). Remarkably, a series of bulky acids were also accepted well by the best mutant Y466A, good yields and D-incorporation being achieved (Fig. 6). This mutant also enabled the potential application of the biocatalytic decarboxylative reaction in the synthesis of complex deuterated compounds from biologically active natural products and drug molecules containing a carboxylic acid functional group, such as Gemfibrozil, Mycophenolic Acid, Zaltoprofen, and Dehydroabietic Acid in good D-incorporation (**5h–5k**). It can also be found that the yields of **5j–5k** is not good probably due to the mutant Y466A can not accept the polycyclic structure well. In addition, the deuterated products achieved with such method could be further transformed into deuterated drug molecules. For example, **5d** could be easily converted into chiral alcohol by ketoreductase[61]. Then, with the known synthetic route, the deuterium could be specifically introduced into (+)-Igmesine (Supplementary Fig. 8)[62].

**Scale-up experiment and $D_2O$ recycling.** This light-driven decarboxylative deuteration method was easily scaled up. Treatment

of 1 g (3.5 mmol) oleic acid (**1D**) with 10 mL DMSO, 40 mL $D_2O$, crude enzyme powder containing about 200 mg $CvFAP$, followed by irradiation for 16 h at 20 °C. Simple extraction and purification steps were then implemented to afford desired product **1d** with 68% yield (567 mg) and 91% D-incorporation.

Furthermore, with a distillation of residual solvent, the recovered $D_2O$ could be used in next reaction without causing a decrease of the D-incorporation of $CvFAP$ (Supplementary Fig. 6).

**Molecular dynamics (MD) simulations.** In order to gain more insights into the differences in the catalytic activity and stereoselectivity of WT $CvFAP$ and the respective mutants in the presence of various substrates, we used MD simulations to model the enzymatic reactions. For medium chain acids and short chain acids, we found that the carboxyl of model substrates remains in close proximity to the N5 atom of FAD, favorable for SET to occur (Fig. 7a–d, Supplementary Figs. 9 and 10). For stereoselectivity, the distance between the carboxyl moiety of (R)-2-(heptan-3-yloxy) acetic acid (**4A**) to the N5 atom in FAD of (R)-selective I398R mutant is clearly shorter than that of the disfavored (S)-isomer (Fig. 7e, f, Supplementary Fig. 11). However, the opposite occurs in G462F (Fig. 7g, h, Supplementary Fig. 12), which is consistent with the reversed (S)-selectivity toward the **4 A** substrate. Moreover, it was observed that the narrow binding tunnel of WT-$CvFAP$ (diameter: 7.1 Å) clearly hinders the access of the bulky substrate, such as 3-phenylbutanoic acid (**5A**) into the proximity of the active site (Fig. 7i). In mutant Y466A, the

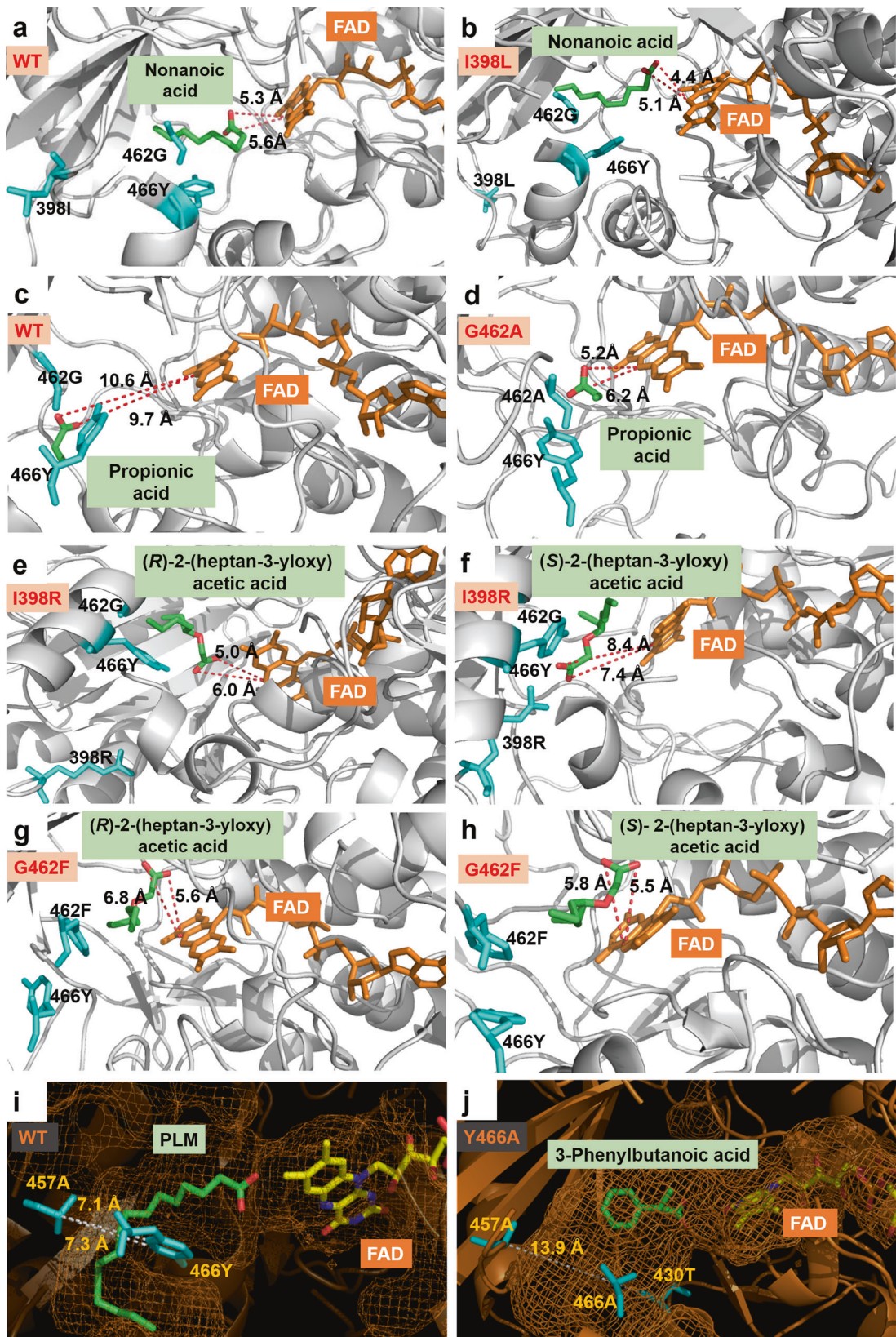

**Fig. 7 MD simulations for gaining insight into the origins of the improved activity and stereoselectivity of CvFAPs. a** and **b**, for nonanoic acid. **c** and **d**, for butyric acid. **e–h**, for 2-(heptan-3-yloxy) acetic acid. **i** and **j**, the change of the binding tunnel's diameter. The docked substrates (green), PLM (palmitic acid, green), FAD (yellow) and some important residues (cyan) are depicted as stick models.

diameter of the binding tunnel is expanded to about 13.9 Å, which is large enough to accept the bulky acid (Fig. 7j, Supplementary Figs. 13 and 14).

In summary, we have demonstrated a photoinduced biocatalytic process for the deuteration of readily accessible carboxylic acids as starting materials and $D_2O$ as the deuterium source. In contrast to traditional photoinduced HAT processes, our method functions well under an air atmosphere without the requirement of thiol as additive. Although the substrate scope of typically biotransformation is narrow, divergent protein engineering was implemented through the FRISM strategy to permit the successful preparation of a diverse range of deuterated compounds in high yields and excellent D-incorporation, and in relevant cases even with good enantioselectivity. Notably, the enantiodivergent directed evolution of *Cv*FAP was also achieved, allowing a series of chiral deuterated products to be obtained by engineered *Cv*FAP with good yields and stereoselectivity. Finally, the scale-up and $D_2O$ recycling experiments were implemented, which demonstrated the practical utility of this biocatalytic process. We anticipate that this ecologically and economically viable synthetic process for the preparation of deuterated compounds will be adopted in both academic and industrial fields.

## Methods

**General information.** All chemical reagents were obtained from commercial sources unless otherwise noted. The $^1H$ and $^{13}C$ NMR spectra were recorded with a Bruker AMX400 MHz spectrometer using TMS as an internal standard. All compounds were characterized by high resolution mass spectra (HRMS) (Waters GCT PremierTM orthogonal acceleration time-of-flight (oa-TOF) mass spectrometry with an EI Source). GC yields and e.r. values were recorded by SHIMADZU™ GCsolution software version 3 in GC-2014 gas chromatography system with Agilent CP-chirasil-Dex CB (FID, $N_2$ as the carrier gas).

**PCR based methods for the construction of *Cv*FAP library.** PCR reactions were performed using WT-FAP plasmid (pET28b) as the DNA template, and forward primers (see Supplementary Table 1) and a silent reverse primer (GATGCCGG-GAGCAGACAAGCCCGTCAGGGCGC). The reaction (50 μL final volume) contained: ddH2O (31 μL), KOD 10X buffer (5 μL), dNTP (5 μL, 2 mM each), forward primers (1.5 μL, 10 μM each), silent reverse primer (1.5 μL, 10 μM each), template plasmid (1.0 μL, 100 ng/μL each), 1 μL of DMSO and 1 μL of KOD-plus-Neo. PCR conditions used were 94 °C, 2 min; 30 cycles of (98 °C, 10 sec; 55 °C, 30 sec; 68 °C, 5 min) and final extension at 68 °C, 10 min. To ensure the elimination of the circular polymethylated template plasmid, 50 μL of PCR reaction mixture was mixed with 2 μL Dpn I (10 U/μL) and incubated overnight at 37 °C, followed by an additional 2 μL of Dpn I for 3.0 h. Upon purification of the Dpn I-digested product with an Omega PCR purification spin column, an aliquot of 20 μL was used to transform 80 μL of *E. coli* BL21 electrocompetent cells. The transformation mixture was incubated with 1 mL of LB medium at 37 °C with shaking, and then spread on LB-agar plates containing Kanamycin (50 μg/mL). A single colony from a plate was incubated in 5 mL of LB medium at 37 °C overnight, and the plasmid was then extracted with an Omega gel extraction column. DNA was sequenced by Sangon Biotech (Shanghai, China). Target mutant was stored with glycerol at −80 °C.

**Protein expression.** 100 μL of stored bacteria was first inoculated in 5 mL of LB medium (containing 50 μg/mL Kanamycin), and was shaken at 200 rpm overnight as preculture. The preculture was used to inoculate large culture (250 mL TB + 50 μg/mL kanamycin in 1 L shake flasks) at 37 °C for about 4 h until OD600 at 0.6–0.8. After cooling at 4 °C for 1 h, 0.2 mM isopropyl β-thiogalactopyranoside (IPTG) was added to induce *Cv*FAP expression. The culture was allowed to express at 20 °C for 24 h with shaking at 200 rpm. Then, cells were harvested by centrifugation at 4500 × *g* and 4 °C for 25 min and the supernatants were discarded. The cells were resuspended in the same buffer (1 g wet cell in 10 mL of phosphate buffer, 50 mM, pH 8.5) and stored at −80 °C. The cells were repeated freezing and thawing for 3 times, and then released the target proteins by sonication. The cell debris were removed by centrifugation at 20,130 × *g* for 15 min at 4 °C, the enzyme solutions were stored at −80 °C for further reaction.

**Preparation of crude enzyme powder.** The frozen crude enzyme solutions were dried in low temperature vacuum for about 10 h. Take care to avoid excessive freeze-drying which probably results in the loss of bound water.

**Protein purification.** The crude enzyme solution was filtered and loaded on a GE Healthcare HisTrap FF Crude column (5 mL) preequilibrated with 50 mM of

phosphate buffer (pH 7.4) containing 0.5 M NaCl and 5 mM imidazole. The enzyme was eluted by 50 mM phosphate buffer with 0.5 M NaCl and 200 mM imidazole. The proteins were dialyzed by 50 mM of phosphate buffer (pH 8.5) for 12 h at 4 °C.

**General procedure for screening.** 0.02 Mmol of substrate was dissolved in 200 μL DMSO, and then was added to 1 mL of crude enzyme solutions of different mutants (1 g wet cell in 10 mL pH 8.5 phosphate buffer). The mixture was shaken at 1500 rpm under the irradiation of Blue LEDs for 12 h at 20 °C, then extracted with ethyl acetate for three times. The yields and e.r. values were determined by gas chromatography.

**Determination of turnover frequency (TOF).** The TOF determination for WT-*Cv*FAP and mutants were performed in phosphate buffer ($H_2O$ or $D_2O$, pH or pD = 8.5) at 1500 rpm and 20 °C under the irradiation of Blue LED in presence of 5–10 mM (excess substrate) substrate and 10 μM FAD-loading enzyme for 1 h (FAD-loading was determined by the absorbance at 460 nm). The TOF was calculated with the formula: $(Conc._{(product)}/Conc._{(FAD-bound\ CvFAP)})$/reaction time.

**General procedure for decarboxylative deuteration.** 4 mL of $D_2O$ (8 mL for medium chain acids) containing substrate (0.40 mmol, dissolved in 1 mL DMSO or CH3CN) and crude enzyme powder (containing *Cv*FAP about 20 mg) was irradiated by 450 nm LED at 20 °C for 12 h. The mixture was then extracted with ethyl acetate for three times. Yields and e.r. values (absolute configurations confirmed by the corresponding *R*-ethers which were synthesized from *R*-alcohols) were determined by GC. Then, the mixture was extracted with DCM and evaporated under reduced pressure, and purified by column chromatography (petroleum ether/ethyl acetate). The D-incorporation was determined by NMR (for products with high boiling points) or HRMS (for products with low boiling points or gaseous products). Note: the D-incorporations of deuterated hydrocarbons with low boiling points were further corrected by comparing their MS data with that of corresponding hydrocarbons without deuterium, due to the EI-MS could generate M-1 peaks for the detection of hydrocarbons with low boiling points (especially gaseous hydrocarbons). With the boiling point of the alkane increases (higher than hexane), the signal of M-1 peaks will disappear (Supplementary Table 2). Moreover, D-incorporations of some volatile deuterated hydrocarbons were also determined by HRMS because low-temperature distillation which was necessary for the concentration of volatile deuterated products during the purification, sometimes could not remove the residual eluent completely and some NMR signals of the residual elution solvents in the range of 0–1 ppm disturbed the accurate determination of D-incorporations by NMR. Alcohols and ketones were performed H-D exchange with 1 M NaOH solution before determined. All D-incorporation determined data were shown in Supplementary Tables (pages 39–103).

**Scaling-up decarboxylative deuteration.** The scale-up reaction was performed as follows: a flask containing 40 mL of $D_2O$ with crude enzyme powder (containing about 200 mg *Cv*FAP) and 1 g (3.5 mmol) oleic acid was shaken at 200 rpm and 20 °C for about 16 h. The mixture was extracted with ethyl acetate for three times. The organic phase was evaporated under reduced pressure, and purified by column chromatography to provide **1d** with 68% yield (567 mg) and 91% D-incorporation.

**General synthesis procedure of α-alkoxy carboxylic acids.** To a solution of sodium hydride (1.20 g, 30 mmol) in dry THF (20 mL) at 0 °C under $N_2$ atmosphere, dry THF (30 mL) containing various alcohols (10 mmol) was added slowly for about 30 min. Then dry THF (30 mL) containing bromoacetic acid (10 mmol) was added dropwise. The reaction was performed under the reflux conditions and monitored by TLC. After completion, 60 mL water was added into the cooled reaction mixture, and 2 M HCl was used to acidize the aqueous layer to pH 2. Hexane and diethyl ether were used to extract the organic layer for three times, respectively. Various pure α-alkoxy carboxylic acids can be obtained after column chromatography on silica gel with the eluent of petroleum ether/ethyl acetate.

**Molecular docking.** Molecular docking calculations were performed by the AutoDock4.2.6 software[63]. After deleting the environmental waters and the sodium counterions, the MD-equilibrated structures of the WT-*Cv*FAP and mutants were set as receptors. Nonanoic acid, butyric acid, 2-(heptan-3-yloxy) acetic acid and benzyl butyrate were set as the substrates with all their rotatable bonds set flexible, respectively. To include all possible binding conformations, a large cubic box comprised of 60 × 60 × 60 girds with the grid spacing of 0.375 Å was used for the docking calculations. Lamarckian genetic algorithm was applied, and each docking calculation contained 150 genetic algorithm runs. The default values were used for all the other parameters. The first conformations of the largest groups in the docking results were selected as the bioactive bindings.

**Molecular dynamics.** MD simulations were performed using the AMBER 12 software[64]. The FF14SB force field was applied for the *Cv*FAP protein[65], the reported parameter for the FAD was used[66], and the restricted electrostatic potential (RESP) atomic charges and the force field parameters generated using the

Antechamber module of AmberTools were applied for different target compounds[67]. The *Cv*FAP proteins and the binding complexes were individually immersed into the center of a truncated octahedron box of TIP3P water molecules with a margin distance of 12.0 Å, $Na^+$ counterions were added by using the AMBER TLEAP module to keep system in electric neutrality[64]. Each system was firstly energy minimized by the steepest descent method for 5000 steps with the protein or binding complex restricted by a harmonic constraint of 100 kcal·mol$^{-1}$Å$^{-2}$. A further conjugate gradient minimization of 5000 steps was performed with no constraint. Then the system was gradually heated from 0 K to 300 K under the NVT ensemble over a period of 500 ps, during which the Langevin thermostat with a coupling coefficient of 1.0 ps and a weak constraint of 10 kcal·mol$^{-1}$Å$^{-2}$ on the protein or binding complex was applied. Each calculation model was then subjected to an equilibrium simulation for 1 ns in order to remove all constraints. The production MD simulations were conducted under NPT ensemble at 300 K for 100 or 150 ns. Periodic boundary conditions were used with a cutoff radius of 12 Å and long-range electrostatic interactions were performed using Particle Mesh Ewald (PME) method[68]. The Berendsen barostat was used to maintain the pressure at 1 bar. The SHAKE algorithm was used to constrain all the covalent bonds involving hydrogen atoms[69]. The time step was set to 2 fs and the individual frames were saved every 10 ps during the production run. All structural figures were prepared using Pymol (http://www.pymol.org/)[70]. All supplementary figures for the comparison of the distance between the carboxyl of substrates and the N5 atom of FAD in variants by MD simulation were prepared using OriginPro (https://www.originlab.com/).

**Reporting summary**. Further information on research design is available in the Nature Research Reporting Summary linked to this article.

## Data availability

The data supporting the findings of this study are available within the paper and its supplementary information files. The source data underlying Figs. 1d–f and h, and Supplementary Figs. 1–7 and Supplementary Table 3 are provided as a source data file. PDB file used in this study is available in Protein Data Bank (PDB) (ID: 5NCC, https://www1.rcsb.org/structure/5NCC). Source data are provided with this paper.

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

## Acknowledgements

This research was funded by National Natural Science Foundation of China (Nos. 91956128, 22111530114), Zhejiang Provincial Natural Science Foundation (No. LY19B020014) and National Key Research and Development Program of China (to Q.W. and J.X.), and by Scientific Research Starting Foundation of Zhejiang University of Technology (No. 2020105009029) (to J.X.). We thank Dr. Jiankai Zou at Analysis and Test Platform of Department of Chemistry, Zhejiang University for assistance during HRMS determination.

## Author contributions

J.X., J.F., Y.L., W.X. and Z.W. contributed equally to this work. J.X. and Q.W. conceived and designed the study. J.X., J.F., Y.L., W.X., D.L. and H.Z. performed the experiments. Z. W. performed the MD simulation. Q.W., J.X. and X.L. analyzed the data. Q.W. and J.X. wrote the manuscript. All authors checked the manuscript.

## Competing interests

The authors declare no competing interests.
