## [Peer Review File · Nature Communications]

Reviewers' Comments:

Reviewer #1:

Remarks to the Author:

The author reported a decarboxylative deuteration of carboxylic acids catalyzed by CvFAP. The method can be applied to long chain acids, and after protein engineering, to medium chain acids, short chain acids, and bulky cyclic acids. To demonstrate the utility of this method, the authors conducted a kinetic resolution reaction to racemic acids for the synthesis of chiral deuterated compounds.

Novelty: Low. This work doesn't really expand our understanding of CvFAP. However it indeed provides a metal-free decarboxylate deuteration methods which is rare from the perspective of synthetic organic chemistry.

Utility: Good. The kinetic resolution part demonstrates the advantage of using enzymes to do this type of deuteration, although this kinetic resolution has been previously reported.

Overall the graphs are carefully made, the tables are organized, and the discussions are clear. I think it is suitable for Nature Communication.

Additional Comments:

There is an example of a prochiral secondary radical being formed in Table 6 (product 5b). I am curious if the trapping of deuterium is enantioselective.

Reviewer #2:

Remarks to the Author:

In this paper, the authors utilise a known photodecarboxylase from *Chlorella variabilis* (CvFAP) to decarboxylate a range of long chain carboxylic acids in D₂O, giving rise to deuterated and variously functionalised alkanes. The authors use mutagenesis to improve the substrate scope of CvFAP to include shorter chain and aromatic carboxylic acids, with rationalisation by molecular modelling.

The CvFAP has received much attention since it was initially reported by Beissen et al. in 2017, and Xu et al. have made valuable contributions previously in developing the enzyme for chiral resolution at scale. The work in the current paper expanding the substrate scope is indeed interesting, but I was not really convinced by the isotopic labelling aspect of the paper, which I don't believe is as significant as the authors describe.

Specific points to address are:

- 1) The authors are not correct to say that "examples of biological deuteration are scarcely developed" (p 2, line 39). There are very many examples of enzymatic/whole cell reactions being performed in D₂O giving rise to deuterated products (often as part of mechanistic studies). The authors give no justification that their reactivity is synthetically significant (e.g. for medchem applications), but rather it just happens to be where the enzyme puts the deuterium.
- 2) Decarboxylation in D₂O is a well-known method of deuteration. There are many examples utilising organic activation (e.g. DOI: 10.1002/anie.201904671), photochemistry (e.g. DOI: 10.1039/c0cc01464h), chemocatalysis (e.g. DOI: 10.1039/b900398c), and even other decarboxylases (e.g. DOI: 10.1007/s10967-009-0012-z). The authors do not cover this literature well.
- 3) Disconcertingly – many of the reactions (e.g. 2c, 3a, 3f - h, 4j, 4k) give %D values well-below 90% (and so could not be explained by, for instance, adventitious H₂O). Where does the H come from?
- 4) The authors switch between MS and NMR to quantify %D incorporation. Whilst this is justifiable in challenging cases – both numbers should be stated in the ESI when possible. How closely do they agree? What is the error in the %D values?
- 5) No error ranges are reported for any result in the paper. For instance, are the KIEs reported in

Figure 1d real or within experimental error?

6) The mutagenesis screenings are carried out in H₂O not D₂O. The authors justify this by referring to a KIE (which seems very small) and expense (but, as the authors acknowledge, D₂O is fairly cheap). As a result, the authors do not explore which mutants give better %D incorporation and why – which is surely critical if the aim of the paper is to prepare an enzyme for selective deuteration? Often, a higher %D incorporation is worth a trade-off of a lower reaction yield.

Personally, I feel the deuteration detracts from an otherwise interesting mutagenesis story. The authors might like to consider separating this work into two manuscripts, and submitting them separately to specialised biocatalysis and isotopic labelling journals, where they would be better suited.

Reviewer #3:

Remarks to the Author:

The manuscript 'Light-driven decarboxylative deuteration enabled by a divergently engineered photodecarboxylase' reports two main items: 1) the concept of decarboxylative deuteration of a range of carboxylic acids and 2) the engineering of the wild-type photodecarboxylase to broaden its substrate scope.

I cannot evaluate the scientific quality of the protein engineering part but it appears to me that this part is largely based on previous methods (reported by Reetz and coworkers) and therefore is possibly not a major breakthrough in the field of protein engineering. The mutants identified, however, convincingly exceed over the wildtype enzyme in terms of activity (and selectivity?). The first part, i.e. decarboxylative deuteration of carboxylic acids to my knowledge represents a novel concept which may find future applications in the fields indicated by the authors. The manuscript clearly demonstrates the feasibility and scope of the approach. I find the analytical data presented convincing and agree with the authors' interpretation.

The section 'Optimization of the reaction conditions.' does not report any optimization but rather some control experiments (most of which had already been reported previously), the proof-of-concept experiment for deuteration. So this section should be renamed and reorganized. I am also a bit astonished why the authors emphasize the fact that the reactions can be conducted under O₂-containing atmosphere. In case of the experiment in the absence of DMSO no deuteration yield is reported. Is that just a lapsus or is DMSO or the cosolvent needed for the deuteration step? I assume it is not, but one never knows...

Apart from these minor things, I enjoyed reading the manuscript and think it deserves publication in Nature Communications.

Response to reviewers' comments

Referee: 1

Comment 1.

The author reported a decarboxylative deuteration of carboxylic acids catalyzed by CvFAP. The method can be applied to long chain acids, and after protein engineering, to medium chain acids, short chain acids, and bulky cyclic acids. To demonstrate the utility of this method, the authors conducted a kinetic resolution reaction to racemic acids for the synthesis of chiral deuterated compounds.

Novelty: Low. This work doesn't really expand our understanding of CvFAP. However it indeed provides a metal-free decarboxylate deuteration methods which is rare from the perspective of synthetic organic chemistry.

Utility: Good. The kinetic resolution part demonstrates the advantage of using enzymes to do this type of deuteration, although this kinetic resolution has been previously reported.

Overall the graphs are carefully made, the tables are organized, and the discussions are clear. I think it is suitable for Nature Communication.

Reply 1:

We thank the reviewer for the positive comments and for considering our manuscript to be important and useful. We agree that this work indeed provides a metal-free decarboxylate deuteration method which is rare from the perspective of synthetic organic chemistry, although it adopts the same mechanism as CvFAP-catalyzed decarboxylation of fatty acids.

Comment 2.

There is an example of a prochiral secondary radical being formed in Table 6 (product 5b). I am curious if the trapping of deuterium is enantioselective.

Reply 2:

Thank you very much for your important comments and suggestions.

We also considered this reaction is an enantioselective proton transfer process. The similar enantioselective process catalyzed by flavin-dependent "ene"-reductase was reported by Prof. Hyster's group¹. Unfortunately, due to the small difference of H and D atom, it is very difficult to determine the e.r. value of product **5b**. We hope further experiments in our lab to trap the alkyl radical with other functional groups will achieve enantioselective results in the future.

Referee: 2

Comment 1.

In this paper, the authors utilize a known photodecarboxylase from *Chlorella variabilis* (CvFAP) to decarboxylate a range of long chain carboxylic acids in D₂O, giving rise to deuterated and variously functionalised alkanes. The authors use mutagenesis to improve the substrate scope of CvFAP to include shorter chain and aromatic carboxylic acids, with rationalisation by molecular modelling.

The CvFAP has received much attention since it was initially reported by Beissen et al. in 2017, and Xu et al. have made valuable contributions previously in developing the enzyme for chiral resolution at scale. The work in the current paper expanding the substrate scope is indeed interesting, but I was not really convinced by the isotopic labelling aspect of the paper, which I don't believe is as significant as the authors describe.

Reply 1:

We thank the reviewer for the comments and for considering our work to be of interest.

We agree that significance of isotopic labelling maybe overstated a bit. Now we have revised the first sentence in Introduction “Deuterium labelling is one of most attractive and powerful tools in the investigation of ...” into “Deuterium labelling is an attractive and powerful tool in the investigation of ...”. Please see it in Page 2, Line 26.

Comment 2.

The authors are not correct to say that “examples of biological deuteration are scarcely developed” (p 2, line 39). There are very many examples of enzymatic/whole cell reactions being performed in D₂O giving rise to deuterated products (often as part of mechanistic studies).

Reply 2:

Thank you very much for your important comment. We agree with it.

The wide use of deuterium labelling in mechanism studies displays the importance of this technology. Therefore, developing some new toolboxes of general and convenient methodology for deuterium labelling is meaningful. Meanwhile, in the examples of mechanistic studies, the product yields and D-incorporations often cannot satisfy the further application. In order to improve the accuracy of description, we changed the sentence “examples of biological deuteration are scarcely developed” into “the toolbox of enzymatic methodologies for deuterium incorporation has remained underdeveloped”. Please see it in Page 2, Line 39-40. Similar revision was also showed in Abstract (Page 1, Line 14).

Comment 3.

The authors give no justification that their reactivity is synthetically significant (e.g. for medchem applications), but rather it just happens to be where the enzyme puts the deuterium.

Reply 3:

Your suggestion is very good for the improvement of this work. Thank you.

We also believe the medchem applications of deuteration methods are very important. For example, herein, the deuterated product **5d** could be asymmetrically reduced by KRED easily². Then, with the known synthetic route, the deuterium could be specifically introduced into (+)-Iglesine³. We added this information in Page 10, Line 158-160, and also in Supplementary Fig. 15.

Supplementary Fig. 15. Derivatization of **5d** to (+)-Iglesine-*d*₁ with reported methods

Comment 4.

Decarboxylation in D₂O is a well-known method of deuteration. There are many examples utilising organic activation (e.g. DOI: 10.1002/anie.201904671), photochemistry (e.g. DOI: 10.1039/c0cc01464h), chemocatalysis (e.g. DOI: 10.1039/b900398c), and even other decarboxylases (e.g. DOI: 10.1007/s10967-009-0012-z). The authors do not cover this literature well.

Reply 4:

Thank you very much for your comments and suggestions. We agree that decarboxylation in D₂O is a well-known method of deuteration, especially for chemical catalysts.

These references you suggested have been added to the manuscript as Ref 12, 17, 18 and 22. And they have been cited at the suitable positions in the Introduction as shown in the following text.

A number of elegant chemical methods for deuteration have been exploited over the past decades⁶⁻¹²... photocatalytic deuteration of synthetically valuable organic molecules has received considerable attentions¹⁵⁻¹⁸...

...the toolbox of enzymatic methodologies for deuterium incorporation has remained underdeveloped²⁰⁻²²...

12. Magro, A. A. N.; Eastham, G. R.; Cole-Hamilton, D. J., Preparation of phenolic compounds by decarboxylation of hydroxybenzoic acids or desulfonation of hydroxybenzenesulfonic acid, catalysed by electron rich palladium complexes. *Dalton Trans.* **24**, 4683-4688 (2009).

17. Patra, T., Mukherjee, S., Ma, J., Strieth-Kalthoff, F. & Glorius, F. Visible-light-photosensitized aryl and alkyl dcarboxylative functionalization reactions. *Angew. Chem. Int. Ed.* **58**, 10514-10520 (2019).

18 Itou, T.; Yoshimi, Y.; Nishikawa, K.; Morita, T.; Okada, Y.; Ichinose, N.; Hatanaka, M., A mild deuterium exchange reaction of free carboxylic acids by photochemical decarboxylation. *Chem. Commun.* **46**, 6177-6179 (2010).

22 Pajak M, Kanska M. Synthesis of isotopomers of L-DOPA and dopamine labeled with hydrogen isotopes in the side chain. *J. Radioanal. Nucl. Chem.* **281**, 365-370 (2009).

Comment 5.

Disconcertingly – many of the reactions (e.g. 2c, 3a, 3f - h, 4j, 4k) give %D values well-below 90% (and so could not be explained by, for instance, adventitious H₂O). Where does the H come from?

Reply 5:

This is a really important comment. And thank you very much for your suggestions.

We also felt very strange for these results after several repetitions. We asked the mass spectrometry staff of our department and they told us the EI mass spectrum could ionize a mount of M-1 peaks for the detection of hydrocarbons with low boiling points (especially gaseous hydrocarbons). Therefore, the D-incorporation data described previously for hydrocarbons with low boiling points were inaccurate. Thus, we rechecked the MS results of all the deuterated hydrocarbons with low boiling points, determined the D-incorporations by comparing with the MS results of hydrocarbons without deuterium, and corrected these results (for **3a-b**, **3f-h**, new %D results were shown in Table 4, and the HRMS spectra of their corresponding hydrocarbons without deuterium were added in Supporting Information for comparison.). With the boiling points of alkanes increase (higher than hexane), such phenomenon of M-1 peaks appearance is eliminated

(M-1 peaks <1%). Typical HRMS results were shown below.

For the compounds **2c** and **4j**, we repeated the experiments and further purified the products to remove the residual compounds which were detected in 0 - 1 ppm with NMR in enzymatic reaction system. These new %D results were shown in Table 3 (**2c**), Table 5 (**4j**), and their new NMR spectra were shown in Page S71-72 and S112-113.

The ratio of M-1 peak to M peak of various hydrocarbons with low boiling points were shown in the table below which has been added as Supplementary Table 2 into Supporting Information.

Supplementary Table 2 The ratio of M-1 peak to M peak of various hydrocarbons with low boiling points

Substrate	HRMS result	
	Ratio _M (%)	Ratio _{M-1} (%)
Ethane	100	66
Propane	100	80
Pentane	100	12
Hexane	100	2
Heptane	100	<1
Octane	100	<1
Decane	100	<1
Undecane	100	<1

The HRMS spectra of various hydrocarbons with low boiling points were shown as follows.

Minimum: 5.00
Maximum: 100.00

Mass	RA	Calc. Mass	mDa	PPM	DBE	i-FIT	Formula
71.0865	12.26	71.0861	0.4	5.6	0.5	2773170.0	C5 IH11
72.0934	100.00	72.0939	-0.5	-6.9	0.0	5546096.5	C5 IH12

Minimum: 1.00
Maximum: 100.00

Mass	RA	Calc. Mass	mDa	PPM	DBE	i-FIT	Formula
85.1021	2.33	85.1017	0.4	4.7	0.5	2774266.3	C6 H13
86.1099	100.00	86.1096	0.3	3.5	0.0	5547151.5	C6 H14

Minimum: 0.50
Maximum: 100.00

Mass	RA	Calc. Mass	mDa	PPM	DBE	i-FIT	Formula
99.9931	5.10	99.9931	0.0	0.0	0.0	2773914.3	C7 H14
100.1247	100.00	100.1252	-0.5	-5.0	0.0	5546811.0	C7 H16

Minimum: 5.00
Maximum: 100.00

Mass	RA	Calc. Mass	mDa	PPM	DBE	i-FIT	Formula
113.1215	5.10	113.1215	0.0	0.0	0.0	5546836.0	C8 H18
114.1399	100.00	114.1409	-1.0	-8.8	0.0	5546836.0	C8 H18

Minimum: 5.00
Maximum: 100.00

Mass	RA	Calc. Mass	mDa	PPM	DBE	i-FIT	Formula
141.1602	5.10	141.1602	0.0	0.0	0.0	5546186.5	C10 H22
142.1720	100.00	142.1722	-0.2	-1.4	0.0	5546186.5	C10 H22

Comment 6.

The authors switch between MS and NMR to quantify %D incorporation. Whilst this is justifiable in challenging cases – both numbers should be stated in the ESI when possible. How closely do they agree? What is the error in the %D values?

Reply 6:

Thank you very much for your important comments.

As the description in Supporting Information (see Page S5): The D-incorporations of products with high boiling points were determined by NMR. Other products with low boiling points or volatile alkanes were determined by HRMS, because the enzymatic reaction system contained some residual compounds which were difficultly removed by low-temperature distillation, while usually showed NMR signals in the range of 0 - 1 ppm and disturbed the accurate determination of D-incorporations by NMR.

We provided both D-incorporation numbers measured by NMR and HRMS for **1a**, **1b** and **1c** in the Supporting Information (see Page S9-S10), and attached the HRMS Spectra of **1a**, **1b** and **1c** in Page S47, S49 and S51. We also compared the difference of D-incorporation determined by NMR and HRMS, respectively, as shown in Supplementary Table 3, the error is acceptable.

Supplementary Table 3 Evaluation of the difference of the D-incorporation determined by NMR and HRMS

Substrate	D-incorporation (%)	
	Determined by HRMS	Determined by NMR
1a	91±1	93±2
1b	94±2	95±2
1c	95±1	96±1

Comment 7.

No error ranges are reported for any result in the paper. For instance, are the KIEs reported in Figure 1d real or within experimental error?

Reply 7:

Thank you very much for your important suggestions.

The Figures 1d, 1e, 1f, 1h showed the average experimental results, now we have added the error ranges in these figures. Please see the revised Fig. 1d-1h. The detailed data were provided in “Source data” file.

Fig.1 | Design of CvFAP-catalyzed decarboxylative deuteration and divergent directed evolution of CvFAP.

Comment 8.

The mutagenesis screenings are carried out in H₂O not D₂O. The authors justify this by referring to a KIE (which seems very small) and expense (but, as the authors acknowledge, D₂O is fairly cheap). As a result, the authors do not explore which mutants give better %D incorporation and why – which is surely critical if the aim of the paper is to prepare an enzyme for selective deuteration? Often, a higher %D incorporation is worth a trade-off of a lower reaction yield.

Reply 8:

Thank you very much for your valuable comments.

According to the reaction mechanism, we considered the major influencing factor for high D-incorporation is the residual H₂O (or the concentration of D₂O) in the reaction system rather than the effect of mutation. Indeed, with the prolongation of freeze-drying time of enzymes and the continuous removal of H₂O, the D- incorporation improved significantly (the results were added in Supplementary Fig. 4). Thus, we chose 10 h freeze-drying time for the treatment of CvFAP before starting reactions. On the other hand, low active mutants also displayed high D- incorporations for the tested substrates (for example, see Table 2: **2a**, **5a**, Table 3: **3a**, Table 4: **3d**, **3e**, Table 5, **4j**) after sufficient lyophilization treatment. Therefore, under the high D- incorporation to be ensured in this experiment, we think the low decarboxylative activity or narrow substrate scope of CvFAP towards some substrates such as medium or short chain acids, racemic acids and bulky cyclic acids, is an indeed bottleneck in the CvFAP-catalyzed decarboxylative deuteration. Thus the aim of directed evolution of this paper is to access different functionalized variants for accepting a diversity of substrates to extend the use of this method. Considering the KIE, operation convenience (without the requirement of freeze-drying of enzymes before starting reactions) and also the experiment cost, we carried out the mutagenesis screenings in H₂O not D₂O.

We have added some corresponding discussion in the text, please see Page 3, Line 71-74, and Page 6, Line 104-106.

Supplementary Fig. 4. The D- incorporation of **1a** and **1c** obtained from WT CvFAP treated with different freeze-drying time.

Comment 9.

Personally, I feel the deuteration detracts from an otherwise interesting mutagenesis story. The authors might like to consider separating this work into two manuscripts, and submitting them separately to specialised biocatalysis and isotopic labelling journals, where they would be better suited.

Reply 9:

We thank the reviewer for considering our work of directed evolution to be of interest. However, both sections of CvFAP-catalyzed decarboxylative deuteration and directed evolution of CvFAP are indivisible. As mentioned in “Reply 8”, the low decarboxylative activity or narrow substrate scope of CvFAP towards some substrates such as medium or short chain acids, racemic acids and bulky cyclic acids, is an indeed bottleneck in the CvFAP-catalyzed decarboxylative deuteration. Thus the aim of directed evolution of this paper is to access different functionalized variants for accepting a diversity of substrates to extend the use of this deuteration method. Actually, the toolbox of enzymatic methodologies for deuterium incorporation has remained really underdeveloped, although the chemical deuteration methods have been well studied.

Referee: 3

Comment 1.

The manuscript ‘Light-driven decarboxylative deuteration enabled by a divergently engineered photodecarboxylase’ reports two main items: 1) the concept of decarboxylative deuteration of a range of carboxylic acids and 2) the engineering of the wild-type photodecarboxylase to broaden its substrate scope.

I cannot evaluate the scientific quality of the protein engineering part but it appears to me that this part is largely based on previous methods (reported by Reetz and coworkers) and therefore is possibly not a major breakthrough in the field of protein engineering. The mutants identified, however, convincingly exceed over the wildtype enzyme in terms of activity (and selectivity?).

The first part, i.e. decarboxylative deuteration of carboxylic acids to my knowledge represents a novel concept which may find future applications in the fields indicated by the authors. The manuscript clearly demonstrates the feasibility and scope of the approach. I find the analytical data presented convincing and agree with the authors’ interpretation.

Reply 1:

We thank reviewer 3 for the nice comment on the enzymatic decarboxylative deuteration approach of carboxylic acids reported in this work, and our mutants identified herein which indeed exceed over the wildtype enzyme in terms of activity and selectivity.

We agree with reviewer 3 that protein engineering part is mainly based on previous methods, FRISM strategy reported by our group and Reetz (see Ref. 43, 44). Besides *Candida antarctica* lipase B (see Ref. 43), this work is another successful application example of FRISM.

Comment 2.

The section ‘Optimization of the reaction conditions.’ does not report any optimization but rather some control experiments (most of which had already been reported previously), the proof-of-concept experiment for deuteration. So this section should be renamed and reorganized.

Reply 2:

Thank you very much for your valuable comments.

We are sorry that some discussion about the optimization of reaction conditions shown in the Supporting Information was omitted in the original version. We have revised this section and added some discussion sentences (Page 3, Line 67-74).

Moreover, we agree with you that the section title should be renamed, and we have changed the section title “Optimization of the reaction conditions” into “Reaction conditions”.

Comment 3.

I am also a bit astonished why the authors emphasize the fact that the reactions can be conducted under O₂-containing atmosphere.

Reply 3:

Thank you for your question. Because the traditional chemical processes of radical decarboxylation need to be performed under N₂ atmosphere to inhibit the radical trapping by O₂. In CvFAP, we supposed the binding pocket offers a relatively inert atmosphere for substrate. Thus, this reaction could proceed under O₂-containing atmosphere. We considered this is also an advantage in contrast with chemical processes.

Comment 4.

In case of the experiment in the absence of DMSO no deuteration yield is reported. Is that just a lapsus or is DMSO or the cosolvent needed for the deuteration step?

Reply 4:

Thank you very much for your careful checking and pointing out this error in the paper.

We are sorry that was a lapsus, we have added the D- incorporation in the absence of DMSO in Table 1.

Comment 5.

Apart from these minor things, I enjoyed reading the manuscript and think it deserves publication in Nature Communications.

Reply 5:

Thank you very much for your very nice comment.

Statement about other revisions:

1. The citation of References in the text were revised because of the renumbered references.
2. The citation of Supplementary Figures in the text were revised because of the renumbered Supplementary Figures.
3. Some new references about “FRISM” were added, see Ref 44, 45.
4. Acknowledgements were revised.
5. The table of content of Supplementary Information has been reorganized according to the requests of journal format.

Answer to Editorial Requests

1. The form of “Editorial policy checklist” has been updated and uploaded.
2. The form of “Reporting summary” has been updated and uploaded.
3. The statement section of “Data availability” in the manuscript has been revised, and the statement of “Source Data” has been added.
4. One source data file has been provided, and the description of this source data files is listed as follows.

File Name: Source data

Description: Detailed data of Fig. 1d-1f and 1h, and Supplementary Figs. 1-6 and Supplementary Table 3.

5. ORCID of corresponding authors have been provided.

References

1. Sandoval, B. A.; Meichan, A. J.; Hyster, T. K., Enantioselective Hydrogen Atom Transfer: Discovery of Catalytic Promiscuity in Flavin-Dependent 'Ene'-Reductases. *J. Am. Chem. Soc.*, **139**,11313-11316 (2017).
2. Xu, J.; Arkin, M.; Peng, Y. Z.; Xu, W. H.; Yu, H. L.; Lin, X. F.; Wu, Q., Enantiocomplementary decarboxylative hydroxylation combining photocatalysis and whole-cell biocatalysis in a one-pot cascade process. *Green Chem* **21**, 1907-1911 (2019).
3. Bagutski V, Elford TG, Aggarwal VK. Synthesis of Highly Enantioenriched C-Tertiary Amines From Boronic Esters: Application to the Synthesis of Igmesine. *Angew. Chem. Int. Edit.* **50**, 1080-1083 (2011).

Reviewers' Comments:

Reviewer #2:

Remarks to the Author:

The authors have made minor revisions in accordance with the reviewer comments. Importantly, the authors have corrected a mistake in the interpretation of the original data-set, which accounts for the low %D values reported for some substrates. I am therefore confident (with some minor points below) that the paper is scientifically sound, but I still have strong reservations that this manuscript meets the threshold of significance required for recommendation for publication in Nature Communications.

Specific comments:

- 1) I am still quite confused about the relationship between %D in the solvent and %D in the product. The authors have included Supplementary Figure 4 – which is helpful, but hard to understand. How does so much H₂O come from the enzyme to give %D levels as low as 10%? A clearer experiment would be to fully dry the enzyme and perform reactions in known mixtures of H₂O and D₂O – does the %D of the product follow the %D of the solvent (as the mechanism suggests it should).
- 2) The revisions explaining the new interpretation of the MS data are not very clear (Especially Lines 77 – 81 of the Supplementary Information).

General comments:

3) I am still not convinced by the novelty of the target deuteration reaction. As I pointed out in the original review (Comment 4), decarboxylative deuteration is already well-known (in D₂O and under mild conditions). Whilst the authors have included the example references I suggested, they have not significantly altered the introduction to address this point in a meaningful way, in fact it reads virtually the same. There are many different types of “deuteration” and the authors do not address their particular area.

4) My point regarding synthetic utility (comment 3 in original review), was not really addressed by the authors either. Other recent deuteration papers in Nature and Nature Comm all achieve challenging and desirable functionalisations with known applications (eg. to inhibit metabolic enzymes, to stabilise chiral centres, to add +4 mass for ADME etc.). The example given by the authors for their catalyst in SI Fig 15 was really not persuasive: the arbitrary target compound is not required for a particular purpose or have any known beneficial properties, and it could be accessed straightforwardly by existing routes.

So really my original review has not changed much. I still find the expansion of the enzyme substrate scope interesting, but the significance of the isotopic labelling is overinflated, especially when compared to other significant advances in the field of deuteration in the last two years.

Reviewer #3:

Remarks to the Author:

From my point-of-view the authors have adequately addressed the issues raised by myself and the other reviewers.

Response to reviewers' comments

Thank you and two reviewers very much for your careful reviewing and important suggestions for this submission. Your comments are extremely valuable and helpful for us to revise and improve our paper. We have studied these comments carefully and have made suitable revisions which we hope meet your approval. The revised text are highlighted in green. We listed all responses to the comments one by one as follows.

Referee: #2

Comment 1.

The authors have made minor revisions in accordance with the reviewer comments. Importantly, the authors have corrected a mistake in the interpretation of the original data-set, which accounts for the low %D values reported for some substrates. I am therefore confident (with some minor points below) that the paper is scientifically sound, but I still have strong reservations that this manuscript meets the threshold of significance required for recommendation for publication in Nature Communications.

Reply 1:

Thank you for your comment.

Comment 2.

Specific comments:

1) I am still quite confused about the relationship between %D in the solvent and %D in the product. The authors have included Supplementary Figure 4 – which is helpful, but hard to understand. How does so much H₂O come from the enzyme to give %D levels as low as 10%? A clearer experiment would be to fully dry the enzyme and perform reactions in known mixtures of H₂O and D₂O – does the %D of the product follow the %D of the solvent (as the mechanism suggests it should).

Reply 2:

Thank you for your comments and suggestions.

Your question is reasonable and I understand. Actually, 10% D-incorporation was obtained only in the case of lyophilization for one hour, because there are still much H₂O in the enzyme sample. As shown in Supplementary Figure 4, with the prolongation of freeze-drying time of enzymes and the continuous removal of H₂O, the D- incorporation improved significantly.

Your suggestion of a clearer experiment is very important, and we have added it as Supplementary Figure 4C. We prepared a fully dried enzyme and performed reactions in known mixtures of H₂O and D₂O with different ratio. As shown Supplementary Figure 4C, the D-incorporation of the product **1a** roughly follows the ratio of D₂O in the solvent mixture.

Comment 3.

2) The revisions explaining the new interpretation of the MS data are not very clear (Especially Lines 77 – 81 of the Supplementary Information).

Reply 3:

We are sorry that our description is not clear enough and we have rewritten these sentences, please see them in Lines 20-24, Page S5 of the Supplementary Information.

Comment 4.

General comments:

3) I am still not convinced by the novelty of the target deuteration reaction. As I pointed out in the original review (Comment 4), decarboxylative deuteration is already well-known (in D₂O and under mild conditions). Whilst the authors have included the example references I suggested, they have not significantly altered the introduction to address this point in a meaningful way, in fact it reads virtually the same. There are many different types of “deuteration” and the authors do not address their particular area.

Reply 4:

Thank you for comments, and we are sorry that our last revisions can't satisfy you. Your suggestion about the introduction is really important and helpful for improving the quality of this paper.

We have rewritten the introduction according to your suggestions. Now we described the advancement of various chemical deuteration and also their limitations. In order to achieve selective incorporation of a single deuterium, various deuterodefunctionalizations were mentioned, especially decarboxylative deuteration was introduced in detail. In the following section of biocatalytic deuteration, biocatalytic reductive deuteration and aromatic L-amino acid decarboxylase-catalyzed deuterodecarboxylations were introduced respectively. Compared with aromatic L-amino acid decarboxylase-catalyzed deuterodecarboxylations, the divergently engineered fatty acid photodecarboxylase-catalyzed deuterodecarboxylations in this work displayed much broader substrate scope. Indeed, biocatalytic aliphatic decarboxylative deuterations providing deuterated alkanes have never been reported.

Actually, the novelty of this work can be highlighted as follow:

1. This is the first example of biocatalytic synthesis of deuterated alkanes. Despite the well-established chemical processes for C-D bond formation, the toolbox of enzymatic methodologies for deuterium incorporation has remained underdeveloped. Biocatalytic reductive deuteration processes only can provide deuterated reduction products such as alcohols or amines. Similarly, aromatic L-amino acid decarboxylase-catalyzed deuterodecarboxylations are also limited within the decarboxylation products of a few aromatic L-amino acids. Moreover, aliphatic deuterated alkanes are very difficult to be prepared both by chemical and biocatalytic deuteration.

2. Divergent protein engineering remarkably expanded the substrate scope of WT-CvFAP. Using specific mutants, several series of substrates including different chain length acids, racemic substrates as well as bulky cyclic acids were successfully converted into the deuterated products (>40 examples). This approach also enabled the enantiocomplementary kinetic resolution of racemic acids to afford chiral deuterated products, which can hardly be accomplished by existing methods.

3. Focused Rational Iterative Site-specific Mutagenesis (FRISM) was successfully applied in the divergent protein engineering of WT-CvFAP, and it does not require the time-consuming and laborious screening.

4. MD simulations explain the results of improved catalytic activity and stereoselectivity of WT CvFAP and mutants.

Comment 5.

4) My point regarding synthetic utility (comment 3 in original review), was not really addressed by the authors either. Other recent deuteration papers in Nature and Nature Comm all achieve challenging and desirable functionalisations with known applications (eg. to inhibit metabolic enzymes, to stabilise chiral centres, to add +4 mass for ADME etc.). The example given by the authors for their catalyst in SI Fig 15 was really not persuasive: the arbitrary target compound is not required for a particular purpose or have any known beneficial properties, and it could be accessed straightforwardly by existing routes.

Reply 5:

Thank you for your comments, although we can't agree with completely.

According to your suggestions in original review, we expand the potential application of the biocatalytic decarboxylative reaction in the synthesis of complex deuterated compounds from biologically active natural products and drug molecules containing a carboxylic acid functional group, such as Gemfibrozil, Mycophenolic Acid, Zaltoprofen, and Dehydroabietic Acid. These decarboxylation products can be obtained in good D-incorporations (**5h-5k**). As new deuterated derivatives of these biologically active molecules, **5h-5k** probably have some special functions or applications in the future. Please see them (**5h-5k**) in Table 6, and also the sentence in Line 4-7 from the bottom, Page 10.

Regarding to deuterated Igmesine shown in SI Fig 15, we agree with your comment that it probably has no desirable functionalisations with known applications (eg. to inhibit metabolic enzymes, to stabilise chiral centres, to add +4 mass for ADME etc.), because this work mainly focused on the methodology of biocatalytic deuteration rather than the properties study of deuterated products. Actually, the study of deuterium drugs is still in its infancy, and the deuterium effect of many drugs remains unknown, especially for the properties of ADME. So it is unreasonable that any new deuterated drug is required to have known applications. In order to make the synthesis route clearer, we have added all steps for the synthesis of Igmesine according to *Angew. Chem. Int. Edit.* 2011, **50**, 1080-1083 (Ref 10 in SI).

Comment 6.

So really my original review has not changed much. I still find the expansion of the enzyme substrate scope interesting, but the significance of the isotopic labelling is overinflated, especially when compared to other significant advances in the field of deuteration in the last two years.

Reply 6:

Thank you for your comments, although we can't agree with completely.

We totally agree that the chemical deuteration methods have been well studied in the past few decades. However, as Prof. Frank Glorius said in (*Angew. Chem. Int. Ed.* 2019, **58**, 10514): “*Despite significant progress in aliphatic decarboxylation, decarboxylative deuteration has remained challenging owing to the requirement of specialized tin hydride or thiol based D-atom donor sources.*” Thus a general protocol for mild decarboxylative deuteration for both aromatic and aliphatic carboxylic acids has remained highly desired. Moreover, as we pointed out in the introduction, the toolbox of enzymatic methodologies for deuterium incorporation has remained underdeveloped, and only biocatalytic reductive deuteration and aromatic L-amino acid decarboxylase-catalyzed deuterodecarboxylations have been reported so far. Indeed, biocatalytic

aliphatic decarboxylative deuterations providing deuterated alkanes have never been reported.

Impressively, this decarboxylation strategy was not limited to aryl carboxylic acids but could also be successfully extended to alkyl carboxylic acids. Despite significant progress in aliphatic decarboxylation, decarboxylative deuteration has remained challenging owing to the requirement of specialized tin hydride or thiol based D-atom donor sources.^[15] Encouraged by this finding, we applied this strategy to

From *Angew. Chem. Int. Ed.* 2019, 58, 10514

As we pointed out in the response to the comment 4, the novelty of this work is remarkable. This is the first report of a general biocatalytic protocol for mild decarboxylative deuteration for both aromatic and aliphatic carboxylic acids. Divergent protein engineering remarkably expanded the substrate scope of WT-CvFAP. Using specific mutants, several series of substrates including different chain length acids, racemic substrates as well as bulky cyclic acids were successfully converted into the deuterated products (>40 examples). This approach also enabled the enantioselective kinetic resolution of racemic acids to afford chiral deuterated products, which can hardly be accomplished by existing methods.

Reviewers' Comments:

Reviewer #2:

Remarks to the Author:

The authors have robustly addressed my previous criticisms and I would now recommend this work for publication in Nature Communications.

In particular, the authors have done a really excellent job on their revision of the introduction. They clearly took my comments on board and have convincingly made a case for the significance of their work in the context of recent developments in isotopic labeling and decarboxylative deuteration. The authors have also comprehensively addressed and clarified the scientific points raised.

Personally, I think a very interesting aspect of the work is the mechanism by which the deuteration occurs, and the authors draw a nice comparison with the Macmillan catalyst of 2017. The incorporation of solvent deuterons at exchangeable residues followed by hydrogen atom/hydride transfer is clearly a strategy which is gaining traction, and the authors may also wish to include references to recent work by Rabinowitz (JACS, 2017, 139, 41, 14368) and Vincent (ACS Catalysis, 11, 5, 2596) in their revised introduction.

This paper will appeal to both mechanistic and synthetic (bio)chemists, and the revisions made by the authors will certainly help to improve the impact of the manuscript with those audiences.

Response to reviewers' comments

Reviewer #2 (Remarks to the Author):

Comment 1.

The authors have robustly addressed my previous criticisms and I would now recommend this work for publication in Nature Communications.

In particular, the authors have done a really excellent job on their revision of the introduction. They clearly took my comments on board and have convincingly made a case for the significance of their work in the context of recent developments in isotopic labeling and decarboxylative deuteration. The authors have also comprehensively addressed and clarified the scientific points raised.

Reply 1:

Thank you for your positive comment.

Comment 2.

Personally, I think a very interesting aspect of the work is the mechanism by which the deuteration occurs, and the authors draw a nice comparison with the Macmillan catalyst of 2017. The incorporation of solvent deuterons at exchangeable residues followed by hydrogen atom/hydride transfer is clearly a strategy which is gaining traction, and the authors may also wish to include references to recent work by Rabinowitz (JACS, 2017, 139, 41, 14368) and Vincent (ACS Catalysis, 11, 5, 2596) in their revised introduction.

This paper will appeal to both mechanistic and synthetic (bio)chemists, and the revisions made by the authors will certainly help to improve the impact of the manuscript with those audiences.

Reply 2:

Thank you for your comments and suggestions.

The references you mentioned have been added as Ref 35 and Ref 38, and the corresponding citations were also added in the Introduction. Please see the Reference section in the manuscript.